# Geospatial Insights into Greece's Desertification Vulnerability: A Composite Indicator Approach

**Kleomenis Kalogeropoulos** [1,*] , **Dimitrios E. Tsesmelis** [2,3] , **Andreas Tsatsaris** [1] , **Efthimios Zervas** [2] , **Christos A. Karavitis** [4] , **Constantia G. Vasilakou** [4] **and Pantelis E. Barouchas** [3]

1   Department of Surveying and Geoinformatics Engineering, University of West Attica, 28 Ag. Spiridonos, 12243 Athens, Greece; atsats@uniwa.gr
2   Laboratory of Technology and Policy of Energy and Environment, School of Applied Arts and Sustainable Design, Hellenic Open University, 26335 Patras, Greece; tsesmelis.dimitrios@ac.eap.gr (D.E.T.); zervas@eap.gr (E.Z.)
3   Department of Agriculture, University of Patras, Messolonghi Campus, 30200 Messolonghi, Greece; pbar@upatras.gr
4   Department of Natural Resources Development & Agricultural Engineering, Agricultural University of Athens, 11855 Athens, Greece; ckaravitis@aua.gr (C.A.K.); vasilakou@aua.gr (C.G.V.)
*   Correspondence: kkalogeropoulos@uniwa.gr

**Abstract:** The Environmentally Sensitive Areas Index (ESAI) is a comprehensive tool for assessing the susceptibility of areas to desertification. This index analyzes various parameters that are vital for environmental health. Through this index, factors such as human activities, geology, soil quality, vegetation and climate patterns are scrutinized. The analysis assigns weights to each participating factor. Thus, the index is derived from the aggregation of four categories (vegetation, climate, soil quality and management practices), and each of them is independently assessed to understand ecological health. In this way, the level of vulnerability to desertification is effectively measured. The application of the index in Greece (for a period of 20 years, 1984–2004) showed signs of environmental degradation and identified many areas with a high risk of desertification. Notably, there was a substantial increase in cultivated land within rural areas, contributing to shifts in the environmental landscape. Furthermore, this period is distinguished as the driest in the last century, with a peak between 1988 and 1993. The consequential rise in irrigation demand, driven by the simultaneous growth of crops and the intensification of agricultural practices, underscores the intricate interplay between human activities and environmental vulnerability.

**Keywords:** vulnerability; spatial analysis; desertification; natural resources management; composite indicators; environmental management

## 1. Introduction

Greece, characterized by its mountainous terrain, exhibits notable elevation variations (0–2918 m Olympus Mountain), giving rise to extensive areas with rugged inclines throughout a significant portion of the country, such as the Pindus Mountain range located in Northern Greece [1–5]. Specifically, gradients exceeding 10% are observed to encompass half of the total land area [2,3,6–10]. These pronounced gradients lead to the forceful discharge of rainwater on surfaces and contribute to severe soil erosion in regions lacking sufficient plant cover [11–13]. These processes stand as primary contributors to the desertification of the country [14–16]. Desertification refers to the process by which fertile land gradually becomes increasingly arid, unproductive, and more similar to a desert environment and it is driven by climate change and human activities [17,18]. It is crucial to differentiate the term "desertification" from the formation of an actual desert [19]. It is a gradual procedure, wherein productive land degrades and progressively becomes inhospitable to vegetation growth, leading to the emergence of barren areas resembling the underlying rock surface [14,20–23].

Desertification remains a contentious subject, characterized by intricate connections between human activities and the natural ecosystem [8,22,24,25]. In previous decades, this challenge primarily stemmed from a lack of consensus on "what to measure" and "how to measure it" [26]. Indicators have emerged as a key approach for representing such complex relationships within the broader system and conveying them to policymakers. These indicators not only monitor the progress of systemic policy objectives but also depict trends and changes in the system's condition [27–30]. The multitude and variety of system interactions are reflected in the diverse array of indicators utilized. Moreover, as new sets of relationships are examined, they necessitate the development of corresponding new indicators [31]. In essence, indicators are increasingly vital tools for conveying information about environmental conditions to decision-makers and the general public [32–36]. Indicators serve the purpose of evaluating environmental performance and assessing alterations resulting from human activities or natural processes [37,38]. Consequently, they prove to be valuable assets in the realm of land management [39,40]. Especially within the environmental sciences, a single indicator often falls short of adequately capturing intricate processes like soil erosion, which stands as a significant catalyst for land desertification. Composite indices, comprised of multiple indicators, offer a more comprehensive approach, enabling the exploration of various avenues in land management and facilitating effective monitoring of environmental conditions [28,41–43].

The decline linked with desertification specifically pertains to the reduction in productivity or the depletion of agricultural and forestry land [44–47]. Desertification is primarily propelled by erosion, posing a critical threat to the deterioration of hilly landscapes. Human activities are a leading factor in expediting this process, frequently accelerating the rate of mechanical soil erosion [48–51]. This heightened erosion, in turn, contributes to the deprivation of the properties of natural resources (physical, chemical, and biological), further exacerbating the vulnerability of these areas. Additionally, the impact is extended to the loss of natural flora, compounding the challenges associated with desertification and emphasizing the need for sustainable land management practices to mitigate these detrimental effects on the environment [52,53].

In the contemporary context, desertification has emerged as a significant and pressing threat, exerting a profound impact on the degradation of the Mediterranean area [2,15,54,55]. This specific area, renowned for its intricate tapestry of diverse ecosystems, cultural richness, and distinct histories of human interaction with the environment, shares common threads that contribute to widespread desertification [14,56,57]. This is driven by various factors, such as diverse climatic conditions characterized by significant fluctuations, frequent and intense rainfall patterns, seasonal droughts, challenging topographical terrain, and an overall limited extent of plant coverage. This complex interplay of environmental variables sets the stage for desertification to occur. Furthermore, the historical trajectory of human intervention in the Mediterranean environment, combined with recent trends of rural abandonment and the resulting reduction in rural potential, magnifies the scope and impact of desertification [20,58–62]. The intricate relationship between human activities and environmental changes has heightened the vulnerability of these regions to desertification, presenting a multifaceted challenge that demands comprehensive understanding and proactive intervention [63,64].

Within the Greek landscape, the severity of desertification is notably pronounced, with various areas confronting a substantial risk [28,65–68]. Regions highly susceptible to desertification encompass regions such as the Peloponnese, Evia, Thessaly, Epirus, and Thrace (Figure 1). The latest studies underscore the urgency of the situation, revealing that 35% of the country is currently at a high risk in terms of desertification, having either undergone or been in the process of desertification, while an additional 49% are considered to be moderately vulnerable to this pervasive environmental challenge [16]. Addressing these complexities requires a holistic approach that integrates environmental management, sustainable land-use practices, and strategic policies to mitigate the impacts of desertification and promote long-term ecological resilience in the Mediterranean region [1–3,69–71].

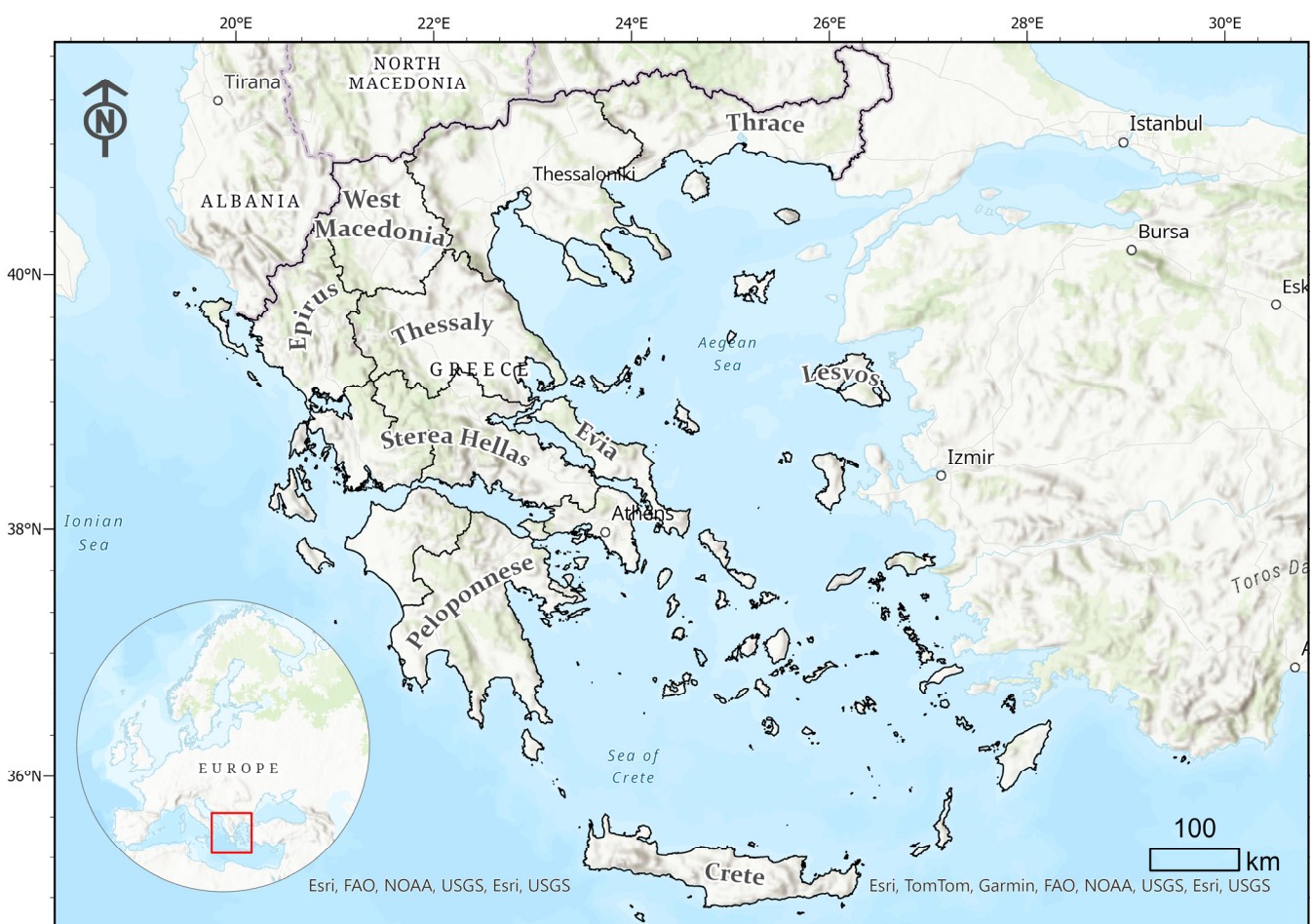

**Figure 1.** The study area.

The Mediterranean climate displays unique characteristics marked by significant seasonal and annual variations in rainfall, high temperatures during the summer season, and extended periods of severe drought [72]. The precipitation patterns, characterized by infrequent yet intense rainfall, along with the region's complex topography featuring steep gradients, frequently result in substantial surface runoff [73]. This phenomenon is accompanied by the loss of fertile soil, significant fluctuations in river runoff, and, at times, catastrophic flooding events [2,74,75]. The Mediterranean's climatic dynamics, shaped by these environmental factors, exert a vital role in influencing the landscape and ecosystems [73,76–79].

Available water is generally insufficient (even with some heavy rainfall) to meet the large needs of vegetation during its growing season. This becomes even more pronounced at certain times [80,81]. This inadequacy in water supply becomes pronounced during severe drought conditions, causing a detrimental impact on the already sparse vegetation in environmentally sensitive areas [78,82–84]. The vulnerability of these regions is exacerbated, as the corrosive effect of rapid rainfall further degrades the compromised vegetation [85–88]. These interconnected environmental processes highlight the intricate balance between natural climatic patterns and the susceptibility of ecosystems to degradation [14,56,57,89]. Understanding and addressing these climatic dynamics is crucial for developing effective strategies to manage and mitigate the risks of desertification in the Mediterranean region [22,62,69,90,91].

In this study, the Environmentally Sensitive Areas Index (ESAI) is used to assess Greece's vulnerability to desertification. The study includes a rigorous validation and calibration process while analyzing the spatial distribution of desertification vulnerabilities.

## 2. Materials and Methods

### 2.1. The Study Area

Greece (Figure 1) is home to a population of 10,482,487 individuals as per the Population Census of 2021 conducted by the Hellenic Statistical Authority, and it covers an expanse of 131,957 square kilometers and boasts a coastline that stretches over 13,676 km [92]. The country experiences a diverse climate, ranging from hot and arid summers to cold and rainy winters. This climatic variety, intertwined with Greece's mountainous terrain and the widespread distribution of its numerous islands, gives rise to a diverse array of microclimates, ecosystems, and landscapes [93–96]. This intricate environmental tapestry not only contributes to the country's natural beauty but also forms the backdrop for a flourishing tourism sector, particularly during the summer season, serving as a pivotal driver for the Greek economy [92,97,98].

Agriculture is the second most significant sector and occupies an expansive 38,540 square kilometers, constituting approximately 20.38% of the total land area. The success of agricultural endeavors is intricately tied to the available water resources in the country, which average around 58 billion cubic meters annually. Despite this seemingly abundant resource, only a mere 12% of this water reservoir is utilized for actual consumption, as indicated by related studies (2008) [99]. On the surface, these figures might suggest Greece is not under immediate threat from water scarcity. However, the country faces challenges due to the inadequate infrastructure for tapping into its considerable surface water potential [93,100,101]. Paradoxically, Greece tends to overexploit its finite groundwater resources, posing a constant risk to their quality. The nation's high vulnerability to water scarcity is highlighted by its dependence on annual rainfall, with instances of shortages recorded in recent decades, including years such as 1989–1890, 1993, 2000, 2003, and 2007 [1,7,9,78]. These episodes' underscore Greece's susceptibility to drought, posing threats to economic stability and leaving the nation exposed to substantial losses, as highlighted by several studies [102].

### 2.2. The Methodology

The beginning of this research focused on Greece's vulnerability to desertification in the period from 1984 to 2004. These provided the impetus for further investigation and analysis of the findings. Initially, the necessary data for the calculation of the Environmentally Sensitive Areas Index (ESAI) were collected. This index demonstrates the vulnerability of an area to desertification using some parameters such as soil characteristics, geology, vegetation cover, climate patterns and anthropogenic activities. Each of these parameters is subjected to a specific categorization with regard to their weighting. Four main quality dimensions are used to calculate the composite indicator (soil, climate, vegetation and applied land management practices). Fifteen sub-indices are generated from these indicators, from which the ESAI is finally calculated. This index is stratified into eight distinct categories (Table 1) and grouped into four types using the geometric mean [42].

**Table 1.** ESAI values, types and categories [42].

| Type | Category | ESAI Values |
|---|---|---|
| Critical | C3 | >1.53 |
| « | C2 | 1.42–1.53 |
| « | C1 | 1.38–1.41 |
| Fragile | F3 | 1.33–1.37 |
| « | F2 | 1.27–1.32 |
| « | F1 | 1.23–1.26 |
| Potential | P | 1.17–1.22 |
| Non affected | N | <1.17 |

The categorization of the ESAI into four types provides a nuanced understanding of vulnerability. Specifically:

Type A refers to areas of severe degradation due to adverse practices, with critical risk to the environment and adjacent areas (e.g., significant erosion as a result of extensive soil loss and runoff during flood events).

Type B refers to areas where a specific modification of the tenuous balance of natural and human activities could lead to desertification, in which case they are classified as sensitive (e.g., a severe drought may lead to the depletion of extensive vegetation, leading to increased erosion and a transition from Type B to Type A).

Type C refers to areas threatened by desertification due to significant changes in various sectors, (e.g., misuse of pesticides, changes in land use practices, changes in social and economic conditions, etc.).

Type D refers to areas that are not vulnerable to desertification.

To calculate the ESAI, data related to soil characteristics, climate patterns, vegetation cover and land management practices are necessary, as shown in Figure 2. This integrated approach aims to provide a comprehensive assessment and categorization of vulnerability, providing a strong basis for informed environmental management and policy decisions.

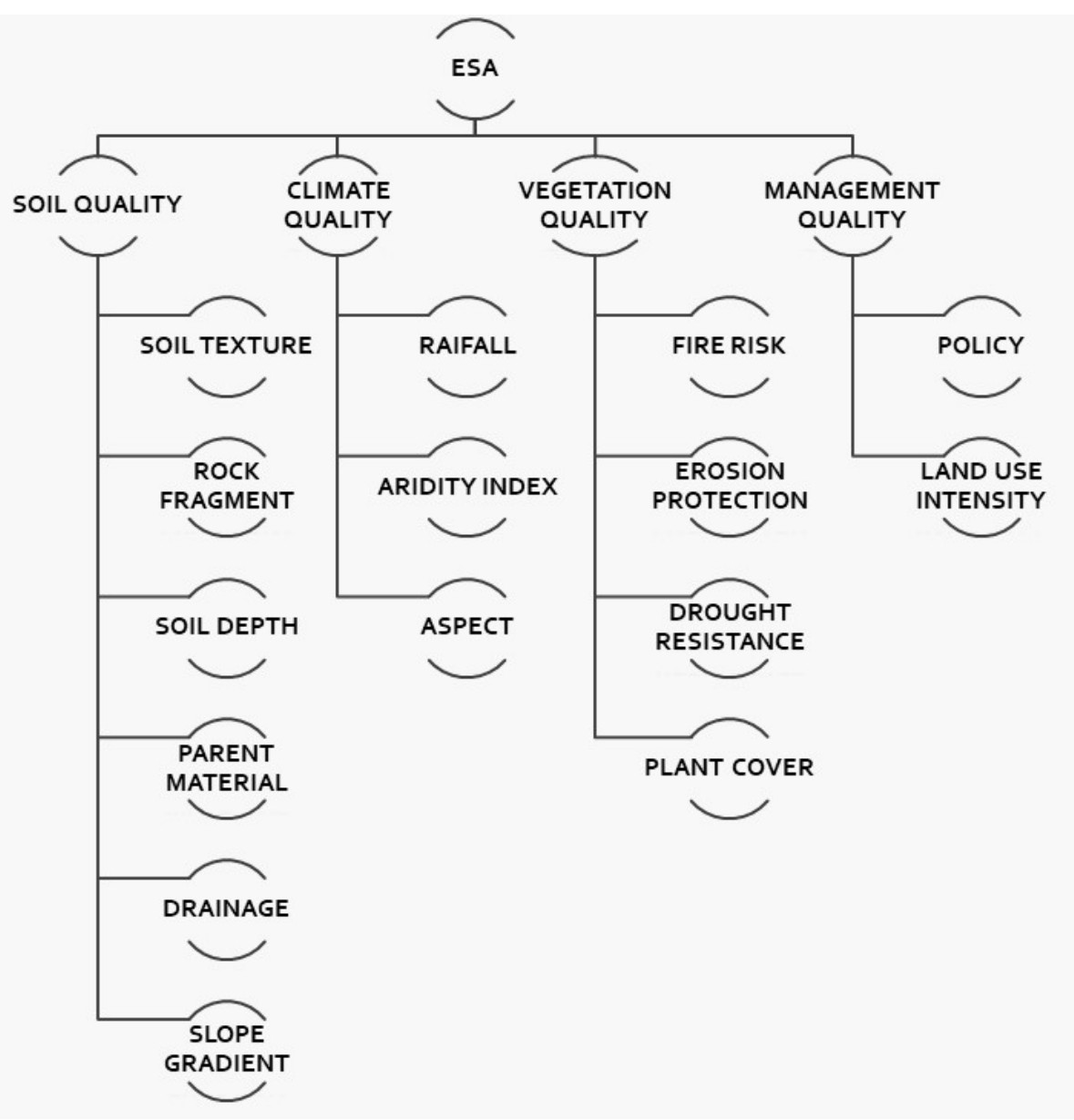

**Figure 2.** ESAI and sub-indices [42].

The above figure presents ESA's Index main indices and sub-indices (Appendix A). In particular ESA is consisting of:

### Soil Quality (SQI).

- Soil Texture. Composition and structure of the soil.
- Rock Fragment. Presence and distribution of rock fragments within the soil.
- Soil Depth. Thickness of the soil layer.
- Parent Material. Geological material from which the soil is derived.
- Drainage. Ability of soil to remove excess water.
- Slope Gradient: Steepness of the terrain.

### Climate Quality (CQI).

- Rainfall. Precipitation distribution.
- Aridity Index. Degree of dryness in the climate.
- Aspect. Direction a slope faces and its impact on microclimates.

### Vegetation Quality (VQI).

- Fire Risk. Likelihood and severity of wildfires.
- Erosion Protection. Effectiveness of vegetation in preventing soil erosion.
- Drought Resistance. Ability of vegetation to withstand periods of water scarcity.
- Plant Cover. Extent and density of vegetation cover the area.

### Management Quality (MQI).

- Policy.
- Land Use Intensity. The degree of human impact on the land through activities such as agriculture or urbanization.

The final step involves ensuring the natural environment works in harmony with key factors like soil health, climate patterns, vegetation cover, and the effectiveness of human efforts to reduce the risk of desertification (referred to as management quality). This critical assessment aims to form a comprehensive understanding of the ecosystems being studied. The delineation of the above types is achieved by applying Equation (1) (the type includes all four indicators). This mathematical expression includes the complex relationships and weighted contributions of each indicator and, therefore, provides a holistic assessment of vulnerability. Equations (1)–(4) provide a classification of areas based on their vulnerability to desertification. This facilitates targeted and evidence-based interventions for sustainable environmental management.

$$SQI = \sqrt[6]{Soil\ Texture \times Rock\ Defragment \times Soil\ Depth \times Parent\ Material \times Drainage \times Slope} \tag{1}$$

$$CQI = \sqrt[4]{Rainfall \times Aridity\ Index \times Soil\ Depth \times Aspect} \tag{2}$$

$$VQI = \sqrt[4]{Fire\ Risk \times Erosion\ Protection \times Drought\ Resistance \times Plant\ Cover} \tag{3}$$

$$MQI = \sqrt{Policy \times Land\ Use\ Intensity} \tag{4}$$

$$ESAI = \sqrt[4]{SQI \times CQI \times VQI \times MQI} \tag{5}$$

Table 1 presents the ESAI values, types, and corresponding categories used for environmental sensitivity assessment. The ESAI is divided into three main types: Critical, Fragile, and Potential, along with a category for Non-affected areas. Under each type, specific categories are defined based on the index values. This classification system facilitates the evaluation and categorization of environmental sensitivity across diverse landscapes.

The methodology was followed according to the criteria and sub-criteria described above. The whole process was carried out in a Geographic Information Systems environment. The software used in this study was ArcGIS Pro ver. 3.2 (ESRI, Redlands, CA, USA). All criteria and sub-criteria formed layers of information in this context, and all the calculations used map algebra functions.

## 3. Results and Discussion

The role of soil is central in arid, semi-arid and dry ecosystems. It is the key factor influencing the productivity and sustainability of terrestrial landscapes. Soil influences biomass production, which is of paramount importance in shaping the overall health and resilience of these environments. Soil quality indicators play a critical role, providing key information on the environmental sensitivity of areas prone to land degradation. It is therefore imperative to map environmentally sensitive areas affected by desertification, as the indicators provide information on water availability and soil erosion resistance as the relationship between the two is determined. The ability of soil to resist erosion further highlights its role in maintaining its structural integrity and fertility. Investigation of soil quality indicators therefore provides a nuanced understanding of the complex interplay between soil characteristics, water dynamics and the wider ecological health of arid and semi-arid regions.

By using the geometric mean equation, a rigorous analysis of the soil was achieved, and in this way, the soil characteristics were understood. The subsequent categorization based on Soil Quality yielded intriguing insights. Remarkably, only a slender 5.22% of the total falls into the premier class, signifying High Quality soil. The majority, a substantial 88.87%, occupies the second class, designating Moderate Quality. Adding a layer of complexity, the third class introduces an additional facet of High Quality, constituting 5.9% of the overall assessment. To visually represent these findings, Figure 3 vividly illustrates that 24.9% of the landscape boasts a high-quality climate scale, depicted in an invigorating shade of green on the map. The moderate climate quality, representing 68.19%, manifests itself in a warm orange hue, while the low-quality areas are visually distinct in red, accounting for 6.91% of the total (Figure 3).

An intriguing revelation emerges when delving into the distribution of low-quality vegetation, which remarkably covers a substantial portion—34.55%, surpassing the percentages associated with both soil and climate quality. Within the intermediate class, an expansive 56.41% is observed, while the high-quality segment occupies a modest 9.03%. This nuanced breakdown underscores the intricate interplay between soil, climate, and vegetation quality, contributing to a comprehensive understanding of the ecological dynamics at play in the assessed regions. Shifting focus to Management Quality, a more favorable scenario unfolds, standing in stark contrast to the other three qualities. In this domain, two distinct classes emerge, with the absence of any representation for low quality. Impressively, the rates for high and average quality stand at 64.06% and 35.94%, respectively. This discrepancy in Management Quality suggests a more positive and effective approach in land management practices, indicating a commendable trend toward higher quality in this critical aspect of environmental stewardship. These findings underscore the need for targeted interventions to enhance soil, climate, and vegetation quality, aligning them with the commendable trends observed in the management of these vital ecosystems. Figure 4 presents the Desertification Vulnerability Maps (spatial distribution for ESAI and averaged ESAI) for the period 1984–2004.

The pervasive desertification in Greece unfolds as a protracted narrative spanning approximately three millennia, punctuated by a concerning decline in both soil productivity and available water reserves. This distressing trend primarily takes root in the olive plant cultivation zones, extending its grip over more than 20% of the nation's total land area. The looming specter of desertification casts a shadow over 30% of Greece's overall expanse, while an additional 49% is teetering on the brink of potential desertification. The geographical areas most acutely under threat include Crete, Lesvos, Eastern "Sterea Hellas", Peloponnese, and specific segments of Thessaly and Thrace. Crucially, the intensification of this environmental crisis across the European Mediterranean region cannot be solely attributed to adverse natural conditions; instead, it intricately intertwines with reckless human actions. Land degradation, a serious threat in Greece and neighboring regions, is becoming a reality in vulnerable areas where natural resources like land, water, and ecosystems have been severely overused. The gradual evolution of this phenomenon

underscores temporal and local disparities, rendering it less immediately perceptible to the societies in question until its irreversible consequences come to the fore. The current predicament has scaled unprecedented heights, with the acceleration of desertification becoming notably pronounced in recent years. Intensified agriculture and overuse of water resources are major drivers behind worsening land degradation in Greece and nearby countries. In grappling with this formidable environmental challenge, a pressing need emerges for urgent and comprehensive measures that transcend national borders. The trajectory towards sustainable environmental practices becomes imperative, necessitating collaborative efforts to mitigate the impact of desertification, restore ecosystems, and safeguard the delicate balance between human activities and the natural environment. Only through concerted actions and a paradigm shift toward sustainable land management can the persistent threat of desertification be effectively curtailed, ensuring the long-term resilience of the affected regions and promoting environmental harmony.

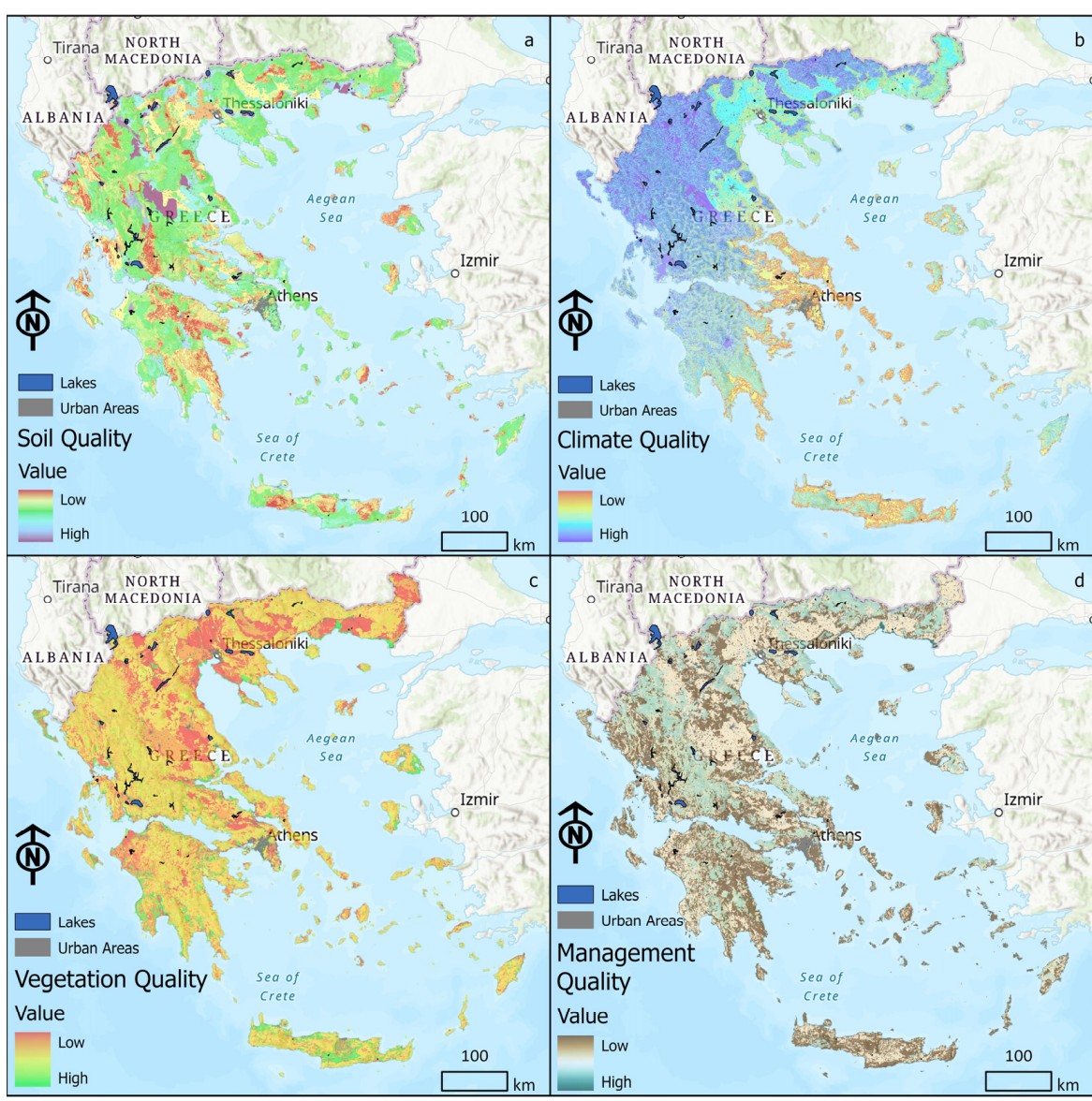

**Figure 3.** Maps of the Sub-Indices of ESAI, (**a**) Soil Quality Index, (**b**) Climate Quality Index, (**c**) Vegetation Quality Index and (**d**) Management Quality Index.

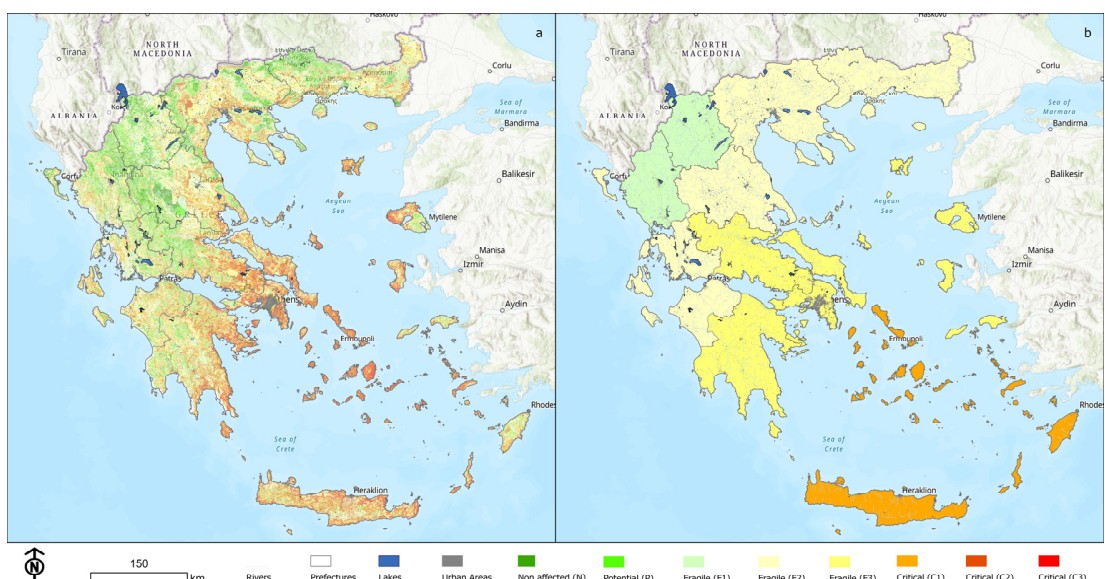

**Figure 4.** Desertification Vulnerability Maps (**a**) spatial distribution for ESAI and (**b**) averaged ESAI based on Prefecture level (period 1984–2004).

Utilizing the quartet of indicators encompassing (Climate, Soil, Vegetation, and Management), relational Equation (4) was systematically employed. The cumulative results derived from this application facilitated the comprehensive assessment of Environmentally Sensitive Areas, specifically those susceptible to desertification. This evaluative process extended into the spatial realm, with the utilization of a Geographic Information System (GIS) environment to generate a corresponding map. Subsequently, the outcomes were meticulously compiled into Table 1, offering a structured presentation of the categorized areas based on their sensitivity to desertification. The categorization unveiled in Table 1 delineates eight distinct classes: Areas deemed non-sensitive to desertification, those classified as potentially sensitive, and a spectrum of sensitivity levels (F1, F2, and F3). Furthermore, the categorization extends to encompass critical areas (C1, C2, and C3), with a particular emphasis on Class C3, where the potential for restoration remains plausible (Figure 5). This nuanced classification scheme not only provides a granular understanding of the diverse environmental sensitivities but also lays the groundwork for targeted interventions and informed environmental management strategies. The amalgamation of relational equations, GIS technology, and systematic categorization enhances our ability to discern and address the intricacies associated with environmentally sensitive areas, fostering a holistic approach towards sustainable land management practices.

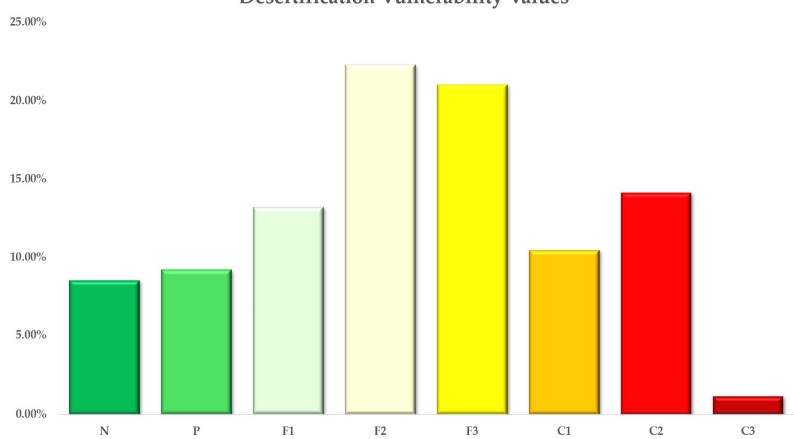

**Figure 5.** Desertification vulnerability percentages for each category.

Upon careful examination of Figure 4a, a discerning pattern emerges in the categorization of environmental sensitivity classes. In the non-affected category (Class 1), there is an observed prevalence rate of 8.52%, denoting areas seemingly immune to the effects under consideration. The potential category (Class 2) follows with a representation of 9.21%, illustrating areas deemed susceptible to potential environmental impact. For the broad spectrum of sensitive areas constituting Classes F1, F2, and F3, the cumulative percentage reaches 56.54%. Delving further, Class F1 occupies 13.21%, Class F2 encompasses 22.31%, and Class F3 accounts for 21.02% of the total, collectively reflecting varied degrees of environmental sensitivity. Moving into the critical areas (Classes C1, C2, and C3), an overall representation of 25.73% is evident. Class C1 occupies 10.45%, indicative of areas teetering on the brink of significant environmental degradation. Class C2, with a percentage of 14.13%, highlights areas with a heightened degree of vulnerability. Notably, Class C3, the most critical category, accounts for 1.14%, underscoring regions where the potential for restoration is notably challenging. This comprehensive breakdown not only provides a quantitative understanding of the distribution of environmental sensitivity classes but also sheds light on the nuanced variations within each category. On the other hand, Figure 4b portrays the averaged value of the index at the prefecture level. The North Aegean and Crete have the highest values of desertification vulnerability. In contrast, the prefectures of West Greece and West Macedonia have the lowest values of desertification.

In order to validate the ESAI, three separate indices were used, namely the Normalized Difference Vegetation Index (NDVI), the Normalized Difference Water Index (NDWI) and the Soil Organic Carbon (SOC). The maps for the NDVI and the NDWI were derived from Landsat 5 (https://developers.google.com/earth-engine/datasets/catalog/landsat-5, accessed on 20 April 2024) for the period 1984–2004 and the map for the SOC was derived from the Soil Grids (https://soilgrids.org/, accessed on 20 April 2024).

The next figure (Figure 6) presents the maps of the used indices.

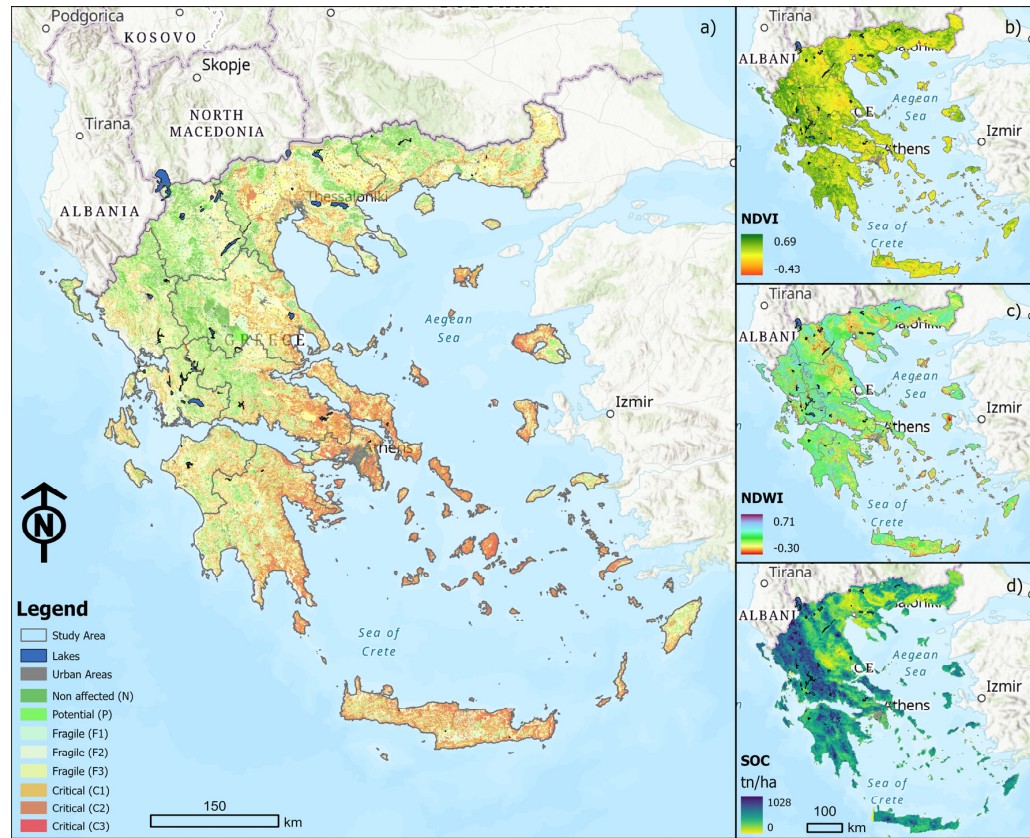

**Figure 6.** The used indices for the validation of ESAI, (**a**) ESAI, (**b**) NDVI, (**c**) NDWI, and (**d**) SOC.

A correlation matrix was created to serve as a tool for understanding the complex interactions between the above indices and their implications for desertification vulnerability in Greece. Table 2 presents this correlation matrix.

**Table 2.** The correlation matrix between ESAI, SOC, NDVI, and NDWI.

| Index | SOC | NDVI | NDWI | ESAI |
|---|---|---|---|---|
| **SOC** | 1 | 0.28 | 0.3 | 0.11 |
| **NDVI** | 0.28 | 1 | 0.54 | −0.13 |
| **NDWI** | 0.3 | 0.54 | 1 | −0.33 |
| **ESAI** | 0.11 | −0.13 | −0.33 | 1 |

The table provides valuable information on the relationships between Soil Organic Carbon (SOC), Normalized Difference Water Index (NDWI), Normalized Difference Vegetation Index (NDVI), and the Environmentally Sensitive Area Index (ESAI). It is understood that this information is crucial for assessing desertification vulnerability in Greece.

The table shows a weak positive correlation (0.11) between SOC and ESAI. This correlation suggests that areas with higher soil organic carbon content might exhibit slightly lower environmental sensitivity. This indicates that soil quality, as represented by SOC, could have a minor influence on the environmental sensitivity of these areas. Furthermore, there is a negative correlation (−0.13) between NDVI and ESAI. This correlation implies that regions with higher vegetation density may tend to be less environmentally sensitive. This suggests that areas with healthier vegetation cover might exhibit lower vulnerability to environmental degradation (which is expected), as indicated by ESAI. Moreover, there is another negative correlation (−0.33) between NDWI and ESAI. This correlation indicates that areas with higher water content, as captured by NDWI, might be less environmentally sensitive. This suggests that water availability, as reflected by NDWI, plays a significant role in determining the environmental sensitivity of an area.

## 4. Conclusions

Desertification has a significant environmental and socio-economic impact on the affected areas, resulting in the chronic degradation of natural resources and the overall productivity of the region. These impacts also extend to the socio-economic sector, with a reduction in rural income and the movement of populations to areas with more favorable employment opportunities. Negative impacts include loss of biodiversity and reduction of agricultural productivity in the soil, affecting local climate systems and freshwater availability. In addition, low-lying areas are exposed to frequent flooding, while the sedimentation of dams reduces their capacity to store water. Socio-economic impacts include a reduction in rural incomes and increased population migration, resulting in increased inequality and reduced livelihood prospects for affected communities. Integrated strategies that address soil degradation, water management and sustainable land use practices are needed to reduce the impacts. Understanding these impacts is critical to formulating effective policies that will enhance resilience, ensure the protection of biodiversity and ensure the survival of affected communities in the face of this challenge.

In this context, indicators offer a means to evaluate desertification vulnerability across various global locations. The methodology derived from these indicators can gauge the efficacy of diverse land management practices, aiding in degradation monitoring and the assessment of desertification combating efforts at both farm and regional levels. By utilizing this developed system of indicators, land users can explore different scenarios to mitigate desertification risk, while also evaluating critical stress factors and their ecosystem-wide impacts. This tool empowers decision-makers worldwide to devise dependable, timely, and effective responses to desertification by estimating how desertification risk evolves with the implementation of relevant management strategies.

The decision support tool provides several advantages for environmental management:

- It allows for the simultaneous demonstration, computation, visualization, and evaluation of numerous desertification indicators.
- It presents resulting desertification risk in a concise, comparable, reproducible, and holistic manner.
- It directly correlates data input with the sensitivity of the output results.
- It incorporates transdisciplinary criteria and evaluation processes, involving experts, administrators, professionals, farmers, and decision-makers, ensuring that input from each group is integral to the tool's successful application.

The ESAI data for Greece shows clear differences between regions. The Cyclades are most at risk, followed by parts of Central Greece, Western Lesvos, and Western Evia. Here, the combined impact of suboptimal soil, vegetation, and climatic qualities, ranging from moderate to low, contributes to the aggravation of desertification. In Central Greece, some areas are particularly vulnerable due to environmental threats (F1, F2, and F3). The quality of the climate emerges as a notable influencing factor, displaying a range from moderate to low quality. Conversely, the soil predominantly exhibits medium quality, with sporadic instances of lower quality. Vegetation quality is relatively low to moderate due to inadequate soil protection against erosion and a limited resistance to drought. Crete grapples with significant challenges, primarily characterized by sensitive areas (F2, F3) and critical zones (C1). The area around Heraklion stands out as particularly problematic, with adverse climatic conditions, poor soil quality, and insufficient vegetation exacerbating the situation. In the Western Peloponnese, overall conditions appear favorable, with minimal vulnerability to desertification. However, the eastern side presents a considerable susceptibility, notably in Skala Laconia, Argolida, and Corinthia, where a pronounced tendency toward desertification is observed. Northern Greece, for the most part, appears less impacted by desertification, with certain exceptions in areas such as the Serres valley, Thessaloniki, Edessa, and the Evros region, where desertification poses a discernible threat.

Finally, the successful development and implementation of decision support systems for mitigating desertification, as well as for natural resources management in general, necessitate adaptable institutions capable of adjusting to the scale and nature of the task. These institutions should be equipped to function effectively in a rapidly changing, evolving, and intricate physical and human environment. To ensure timely and efficient implementation, the outcomes produced by these systems should be integrated into a policy framework that embraces a structured participatory approach involving stakeholders, experts, end-users, and decision-makers.

**Author Contributions:** Conceptualization, D.E.T., K.K., C.A.K. and A.T.; methodology, D.E.T., K.K., P.E.B., A.T., E.Z., C.A.K., C.G.V. and P.E.B.; formal analysis, D.E.T., K.K. and C.G.V.; data curation, D.E.T., K.K. and C.G.V.; writing—original draft preparation, D.E.T., K.K. and A.T.; writing—review and editing, D.E.T., K.K., P.E.B., A.T., E.Z., C.A.K., C.G.V. and P.E.B.; visualization, D.E.T., K.K. and C.G.V. All authors have read and agreed to the published version of the manuscript.

**Funding:** This research received no external funding.

**Data Availability Statement:** Data are contained within the article.

**Conflicts of Interest:** The authors declare no conflicts of interests.

**Appendix A**

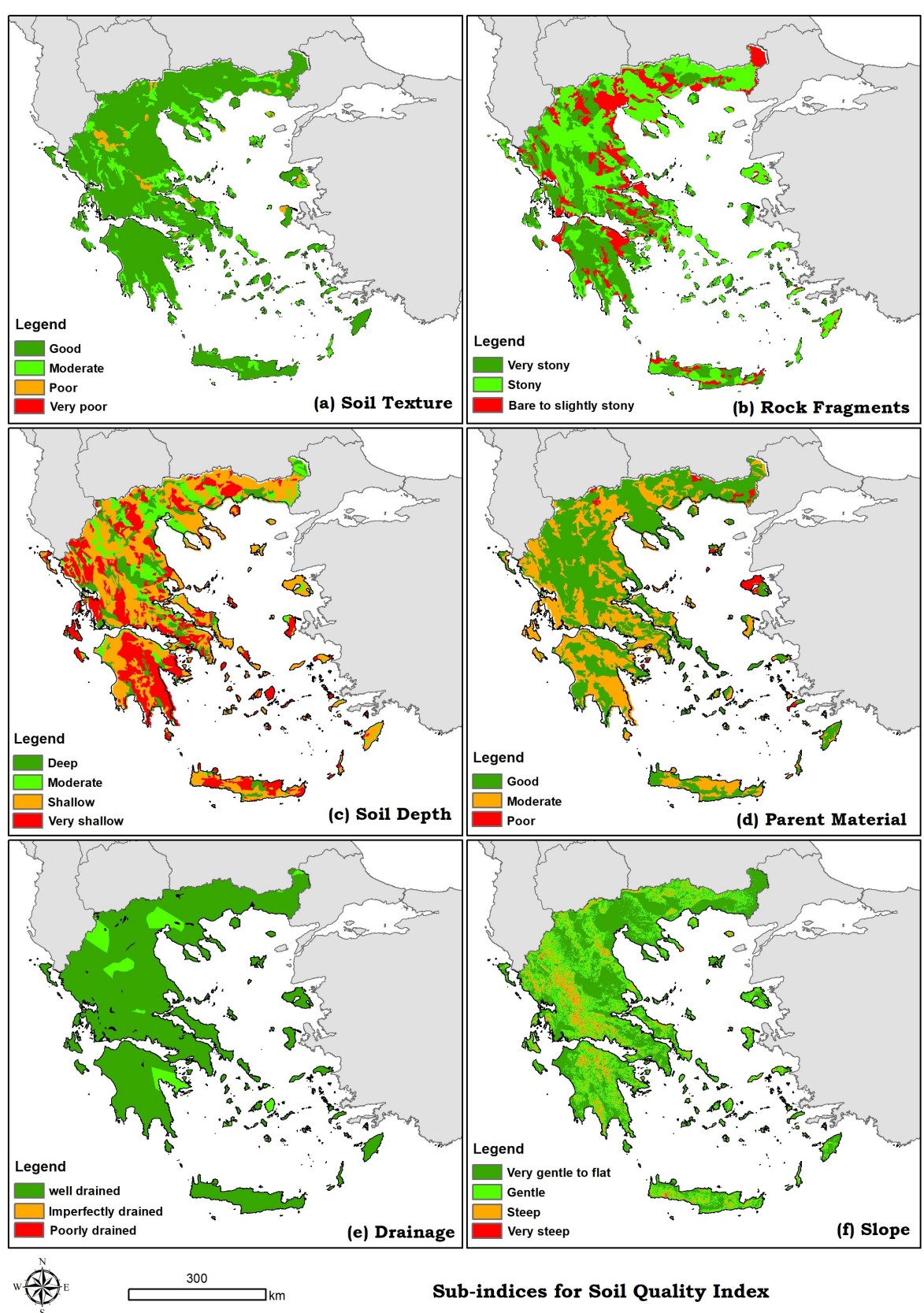

**Figure A1.** Sub-indices for Soil Quality Index.

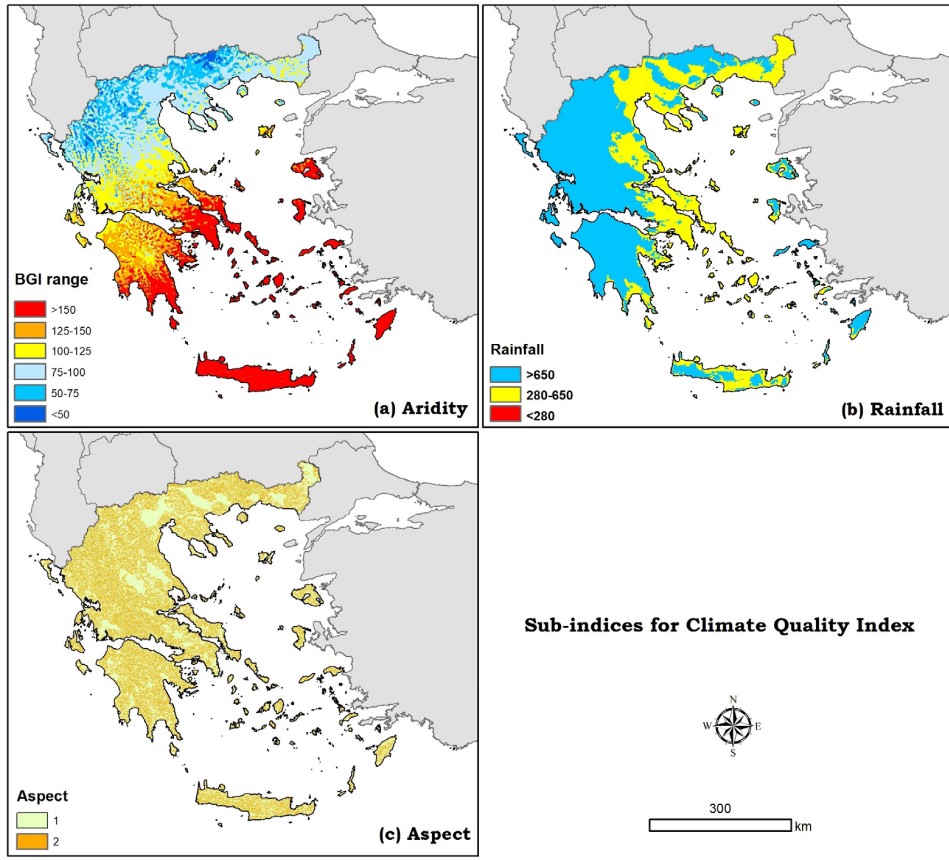

**Figure A2.** Sub-indices for Climate Quality Index.

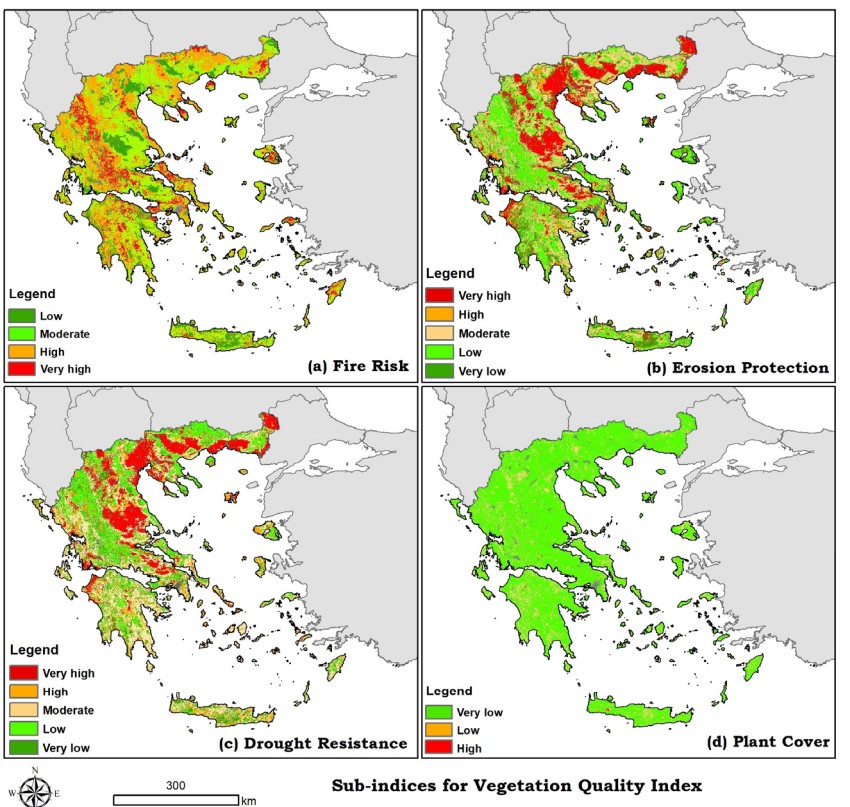

**Figure A3.** Sub-indices for Vegetation Quality Index.

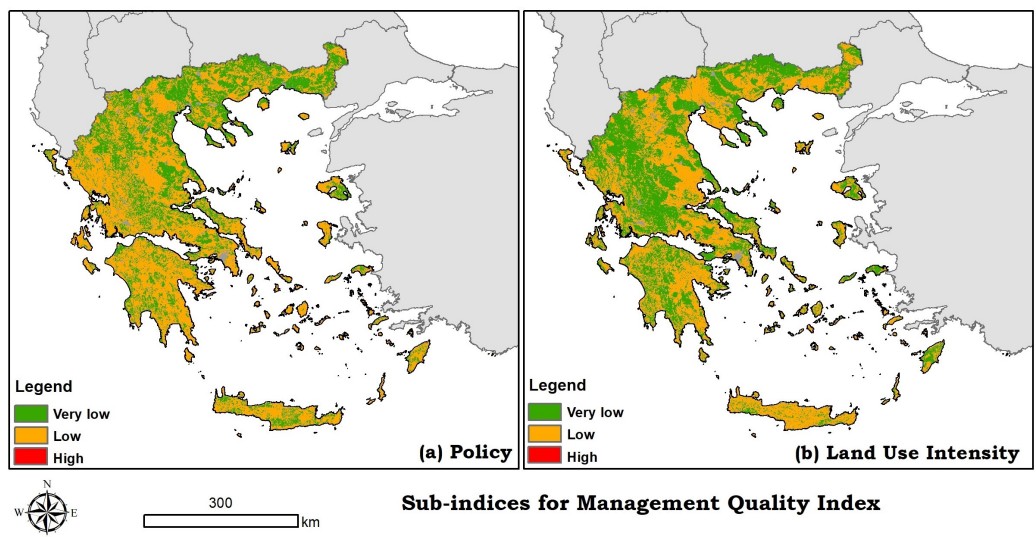

**Figure A4.** Sub-indices for Management Quality Index.

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
