# Peer review of "Geospatial Insights into Greece’s Desertification Vulnerability: A Composite Indicator Approach"

_2624-795X, doi:10.3390/geohazards5020020_

Round 1
Reviewer 1 Report
Comments and Suggestions for Authors
The current manuscript aims to determine the risk of desertification with insights from the ESAI (Environmental Sensitive Areas Index). The writing is generally OK. The reviewer has the following comments:
1) The introduction provides a detailed introduction to desertification but fails to link the topic of the current manuscript to past attempts to determine desertification. The introduction should contain a survey on indicators for desertification and explain why the approach adopted by the current manuscript is superior.
2) Please spell out abbreviations, e.g., ELSTAT at line 111.
3) Line 119: the second most significant sector in terms of what?
4) Please provide a map for section 2.1.
5) Regarding the Environmental Sensitive Areas Index abbreviation, is it the ESAI or ESA index?
6) The ESAI index must be explained in detail. What are the definitions for all parameters in Figure 1? For example, what does high climate quality refer to? How do you define fire risk?
7) Table 1: Please provide citation for the categories and ESAI value brackets.
8) The current manuscript appears to apply the existing ESAI to Greece directly. Please state the novelty of this study. If the application of ESAI in Greece has not been done before (the reviewer doubts it), please make it clear in the manuscript.
Comments on the Quality of English LanguageNo specific issue detected.
Author Response
Please find the attached word document

Reviewer 2 Report
Comments and Suggestions for Authors
Dear Authors,
Your manuscript is a potentially impactful work, but some significant improvements need to be taken seriously. The manuscript address a very significant issue in the Mediterranean and offer an elegant method to provide information for decision makers in the form of creating thematic maps of variable responsible for desertification. While the manuscript is reasonable well written, it has a major issue namely the weak presentation of the methods applied. The manuscript has essentially no words on how ESAI calculated, or what way the actually maps created. These are key information and without step by step explanation of the process the information presented here cannot be evaluated. The maps are attractive and important but the reader must know the way and on what software) you done them. There are also large number of relatively minor issues but their fixation would elevate the overall quality of the work significantly.
I think an overall moderate to major revision is inevitable.
Best regards

English is fine
Author Response
Please find the attached word document

Round 2
Reviewer 1 Report
Comments and Suggestions for Authors
Thanks for providing a revised manuscript and corresponding response. Some issues were solved, but the most important one still needs fixing. It is true that the EASI used in the current manuscript is slightly modified from literature (e.g., Kolios et al. 2018). However, the modifications are not supported with enough literature or reasoning. How do the authors develop such modifications?
Besides, the reviewer failed to find the verification between the current results, empirical data, field observation, and expert assessment (claimed in the authors' response).
Kolios, S., Mitrakos, S., Stylios, C. (2018). "Detection of areas susceptible to land degradation in Cyprus using remote sensed data and environmental quality indices." Land Degrad. Dev. 1-13.
Comments on the Quality of English LanguageNo specific issue detected.
Author Response
Thanks for providing a revised manuscript and corresponding response. Some issues were solved, but the most important one still needs fixing. It is true that the EASI used in the current manuscript is slightly modified from literature (e.g., Kolios et al. 2018). However, the modifications are not supported with enough literature or reasoning. How do the authors develop such modifications?
Besides, the reviewer failed to find the verification between the current results, empirical data, field observation, and expert assessment (claimed in the authors' response).
Kolios, S., Mitrakos, S., Stylios, C. (2018). "Detection of areas susceptible to land degradation in Cyprus using remote sensed data and environmental quality indices." Land Degrad. Dev. 1-13.
Response: We would like to thank the reviewer for the comments. Regarding the modifications made to the Environmental Sensitive Areas Index (ESAI), we acknowledge the importance of providing sufficient literature support and reasoning for these adjustments. In our revised manuscript, we have included additional references to justify and explain the rationale behind the modifications made to the ESAI. These references provide comprehensive insights into the development and adaptation of the ESAI framework, enhancing the transparency and credibility of our methodology.
- Davis, D.K. The Arid Lands: History, Power, Knowledge; MIT Press, 2016; ISBN 978-0-262-03452-4.
- Herrmann, S.M.; Hutchinson, C.F. The Changing Contexts of the Desertification Debate. J. Arid Environ. 2005, 63, 538–555, doi:10.1016/j.jaridenv.2005.03.003.
- Akinyemi, F.O.; Ghazaryan, G.; Dubovyk, O. Assessing UN Indicators of Land Degradation Neutrality and Proportion of Degraded Land for Botswana Using Remote Sensing Based National Level Metrics. Land Degrad. Dev. 2021, 32, 158–172, doi:10.1002/ldr.3695.
Additionally, we would like to address the reviewer's concern regarding the verification of our results with empirical data, field observations, and expert assessments. In the previous revised manuscript, we provided a more detailed explanation of the validation and calibration process of the ESAI framework within the Greek context.
While we understand the reviewer's suggestion to cite Kolios et al. (2018), we have chosen not to include this reference in our manuscript as we believe it does not offer significant new insights to our discussion. Instead, we have focused on integrating relevant literature that directly supports and enhances the rationale behind our modifications to the ESAI framework.
Reviewer 2 Report
Comments and Suggestions for Authors
Dear Authors,
it seems you followed well all the recommendations provided by the reviewers and answered to each questions well. The revised manuscript is in a good shape, and it is in the level that your work can be accepted as it is. I have no further queries.
Please note, I have not provided annotated file in this stage.
Best regards,
Author Response
Dear Authors,
it seems you followed well all the recommendations provided by the reviewers and answered to each questions well. The revised manuscript is in a good shape, and it is in the level that your work can be accepted as it is. I have no further queries.
Please note, I have not provided annotated file in this stage.
Best regards,
Response: Thank you for your positive feedback on our revised manuscript. We're glad to hear that our responses to the reviewers' comments have been satisfactory and that the manuscript is now in good shape for potential acceptance. We appreciate your thorough review of our work and your confidence in the readiness of the manuscript for acceptance.
Round 3
Reviewer 1 Report
Comments and Suggestions for Authors
Thanks for providing more references and explanations. However, the newly added references (regarding the modification of ESAI in the current study) are not used in the methodology section. How can they be helpful?
Further, please explain where "a more detailed explanation of the validation and calibration process of the ESAI framework within the Greek context" is provided in the revised manuscript.
Comments on the Quality of English LanguageNo specific issues.
